# Q-MAP: A CONVOLUTIONAL APPROACH FOR GOAL-ORIENTED REINFORCEMENT LEARNING

## ABSTRACT

Goal-oriented learning has become a core concept in reinforcement learning (RL), extending the reward signal as a sole way to define tasks. However, as parameterizing value functions with goals increases the learning complexity, efficiently reusing past experience to update estimates towards several goals at once becomes desirable but usually requires independent updates per goal. Considering that a significant number of RL environments can support spatial coordinates as goals, such as on-screen location of the character in ATARI or SNES games, we propose a novel goal-oriented agent called Q-map that utilizes an autoencoder-like neural network to predict the minimum number of steps towards each coordinate in a single forward pass. This architecture is similar to Horde with parameter sharing and allows the agent to discover correlations between visual patterns and navigation. For example learning how to use a ladder in a game could be transferred to other ladders later. We show how this network can be efficiently trained with a 3D variant of Q-learning to update the estimates towards all goals at once. While the Q-map agent could be used for a wide range of applications, we propose a novel exploration mechanism in place of $\varepsilon$-greedy that relies on goal selection at a desired distance followed by several steps taken towards it, allowing long and coherent exploratory steps in the environment. We demonstrate the accuracy and generalization qualities of the Q-map agent on a grid-world environment and then demonstrate the efficiency of the proposed exploration mechanism on the notoriously difficult Montezuma's Revenge and Super Mario All-Stars games.

## 1 INTRODUCTION

In the reinforcement learning (RL) setting, an agent attempts to discover and learn to solve a particular task in the environment. The task is generally specified in the form of a reward signal that the agent strives to maximize. While such reward signal is sufficient to describe any task in the environment, it can be advantageous to provide the agent with the task description in form of a specific goal to reach. These goals could be provided by a designed curriculum, or generated by a different agent and have shown to be advantageous for exploration in sparse-reward tasks.

General value functions (GVFs) formalize the concept of goal-oriented RL by creating a specific value function per goal (Sutton et al., 2011), which can be combined in a fixed array (also called Horde) of independent daemons, each simultaneously trained off-policy. Universal value function approximators (UVFAs) extend GVFs by utilizing a single value function that is provided with the goal specification in input, enabling it to generalize between goals. One of the main challenges in such goal-parameterized policies is to make the best use of the data collected so far, to generalize to goals that were not necessarily selected when the data were collected. Training off-policy, using random goals and bootstrapping from the estimated value of next states allows in theory to learn to reach all goals simultaneously. With the Horde framework, this requires to update separately each of the daemons using a single transition while with UVFA, each goal needs to be passed in input to the same policy independently. Because the number of goals can be very large, possibly infinite, both of these approaches quickly become intractable if one wishes to update simultaneously towards all goals using a single transition.

While goal specification can be very complex, for many RL environments, simple spacial coordinates are sufficient to describe many tasks of interest. ATARI or NES games are a great example of

such environments, where the agent's location is often easily identifiable, and thus the goal can be defined in terms of on-screen coordinates. These goals are likely to be highly correlated both to the neighbouring goals and to the visual surroundings. Our first contribution is an agent, called Q-map, that uses a convolutional autoencoder-like architecture for goal-oriented RL, that can be used to efficiently and simultaneously (in a single forward pass) produce value estimates for an entire range of possible goals in compatible environments. We describe how to efficiently train such an agent and show that such approach is able to generalize to unseen environments to a good degree.

As a demonstration how Q-maps can be used, our second contribution is an exploration algorithm that exploits the capacity to extract at the same time an estimated distance towards all goals and a policy to reach one selected randomly at a desired distance. We demonstrate how the $\varepsilon$-greedy exploration mechanism in DQN (Mnih et al., 2015) can be replaced with random goals reaching, expanding the exploration boundaries much further, especially in environments where actions can cancel each other or lead to terminal states. The source code and several videos can be found at https://sites.google.com/view/q-map-rl.

## 2 Q-MAP

### 2.1 BACKGROUND

We consider the standard RL framework (Sutton and Barto, 1998), in which an agent interacts sequentially with its environment, formalized as a Markov Decision Process (MDP) with state space $\mathcal{S}$, action space $\mathcal{A}$, reward function $r(s, a, s')$ and state-transition function $p(s'|s, a)$. At each time step $t$, the agent provides an action $a_t$ based on the current state $s_t$ and sampled from its policy $\pi(a_t|s_t)$. The environment responds by providing a new state $s_{t+1}$ and a reward $r_{t+1}$. Some states can be terminal, meaning that no more interaction is possible after reaching them, which can be simply considered as a deadlock state that only transition to itself, providing no reward.

The action-value function of the policy $Q^\pi(s, a) = \mathbb{E}_{s' \sim p(s'|s,a), a' \sim \pi(a'|s')}\left[r(s, a, s') + \gamma Q^\pi(s', a')\right]$ indicates the quality (Q-value) of each possible immediate action when following the policy afterwards. In the Q-learning algorithm (Watkins and Dayan, 1992), the action-value function of the optimal policy $\pi^*$ is iteratively approximated by updating the estimated Q-values $Q_{t+1}(s, a) \leftarrow (1 - \alpha)Q_t(s, a) + \alpha\left(r + \gamma \max_{a'} Q_t(s', a')\right)$ using previously experienced transitions $(s, a, s', r)$ and a learning rate $\alpha$. At each time step, this learned action-value function can be used to take greedy actions $a = \arg\max_a Q(s, a)$ or random actions uniformly $a \sim \mathcal{U}(\mathcal{A})$ with probability $\varepsilon$. This basic exploration method is called $\varepsilon$-greedy and using the estimate of the value at the next step to improve the estimate at the current step is known as bootstrapping. Finally, the fact that the target $r + \gamma \max_{a'} Q_t(s', a')$ does not rely on the policy used to generate the data, allows Q-learning to learn off-policy, efficiently re-using previous transitions stored in a replay buffer or generated by a another mechanism.

While an action-value function is usually specific to the rewards defining the task, the GVFs $Q_g^\pi(s, a)$ are trained with pseudo-reward functions $r_g(s, a, s')$ that are specific to each goal $g$. The Horde architecture combines a large number of independent GVFs trained to predict the effect of actions on sensor measurements and can be simultaneously trained off-policy. UVFAs extend the concept of GVFs by adopting a unique action-value function $Q^\pi(s, a, g)$ parameterized by goals and states together, enabling interpolation and extrapolation between goals.

It is possible to extract a notion of the distance $\gamma^{k-1}$ between the current state and the desired goal in terms of the number of actions $k$ by training a GVF- or UVFA-based agent with a pseudo-reward of 1 and termination in case the goal is reached, or otherwise rewarding 0. While the agent could in theory learn to reach any goal by attempting to reach them one at a time, the process can become very slow with increasing number of possible goals. However it is theoretically possible to use a single transition to update off-policy the Q-value estimates for the entire range of available goals. In hindsight experience replay (HER) (Andrychowicz et al., 2017), the goals used for the updates are chosen in the episode that gave a transition $(s_t, a_t, s_{t+1})$, after the step $t$ while in Schaul et al. (2015a), the goals are simply chosen randomly, as updating for all becomes computationally difficult, requiring two forward passes through the network for every goal update.

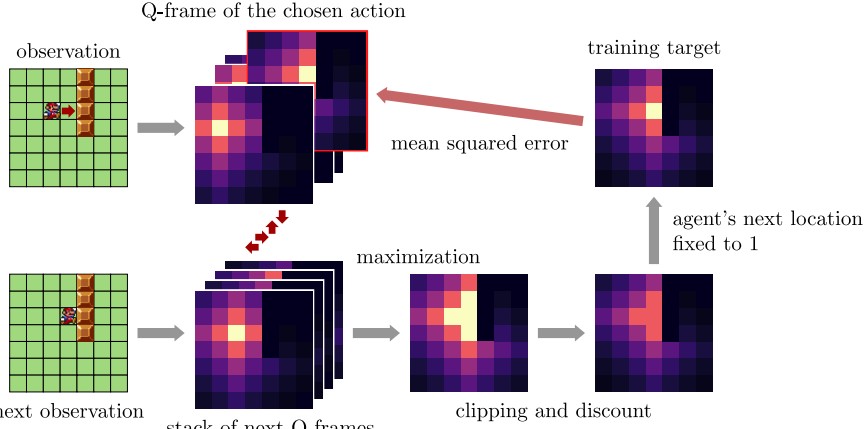

Figure 1: Training process for a Q-map agent updating the prediction towards all goals at once.

## 2.2 THE PROPOSED AGENT

The core concept of the proposed agent called Q-map is to use a convolutional auto-encoder-like neural network to simultaneously represent the GVFs of all possible on-screen locations for all possible actions from the raw screen pixels. This allows to efficiently share weights and to identify correlations between visual patterns such as walls, enemies or ladders and patterns in the distance towards on-screen locations. This means, for example, that learning how to navigate ladders can be transferred to other ladders later in a game. While a stack of greyscale frames are given in input, a stack of 2D *Q-frames* are produced in output where a single voxel at column $x$, row $y$ and depth $a$ represents the expected distance to the on-screen coordinates $(x, y)$ given action $a$.

A great advantage of the Q-map architecture is that it allows one to query the estimated distance towards all goals in a single pass through the network and to very efficiently create a learning target from these estimates with just an additional pass using a 3D version of the Q-learning algorithm. Rewards are matrices comprising of zeros with only the value at the next coordinates set at 1 and Q-values are the 3D tensors consisting of a stack of 2D Q-frames with the maximization of Q-values performed on the depth axis and no bootstrapping at the new location. Because the Q-values are bounded to $[0, 1]$, clipping of the estimated Q-frames at the next state can be performed to speedup and stabilize the learning. Creating the target for an update thus consists of five steps: 1. perform a forward pass through the network with the next observation 2. clip and discount the values 3. maximize through the depth axis 4. replace the value at the next coordinate with 1 or in the case of a terminal state, use a zero matrix with the 1 at the new location 5. offset the frame by the required number of pixels if the observations involve a sliding window that moved during the transition. It is worth noting that a similar convolutional structure was used by Jaderberg et al. (2016) to represent Q-values for the pixel-control auxiliary task of the UNREAL agent. However the architecture was not considered for a goal-oriented framework, with a reward function based on the change in pixel values, without terminal transition at every step.

Q-maps are particularly suited for environments in which it is possible to locate the agent's position in screen coordinates, which could either be provided by the environment (e.g. from the RAM in video games), or a classifier trained to localize the agent on the screen. While these coordinates are used during the training to create the targets for the Q-learning, it does not however preclude one from only using raw frames as input for the Q-map agent. In some games, such as Super Mario Bros. (used later in the experiments), the screen scrolls while the player's avatar moves, and thus only a portion of an entire level is shown at once. In the proposed Q-map implementation, we chose to use the coordinates available on the screen as possible goals and not coordinates over an entire level, thus the map is local to the area around the agent. Finally, while the distance to the goals could more directly be represented by the expected number of steps, the decay factor $\gamma$ forces the values to be bounded to $[0, 1]$, allowing the value of unreachable coordinates to naturally decay to 0 and the coordinates one-step away to have a value of 1.

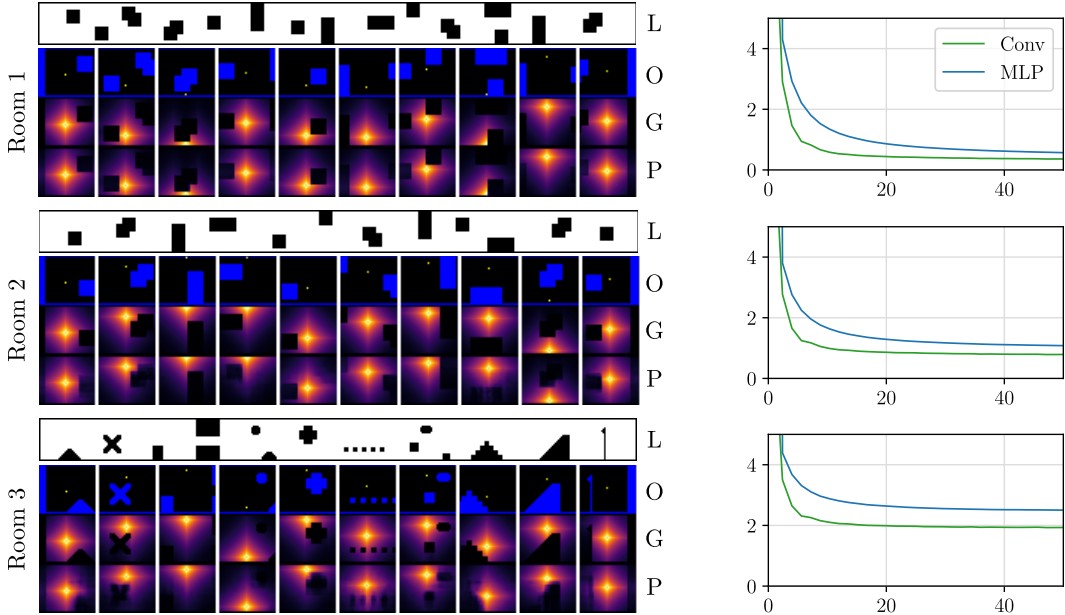

Figure 2: Left: For each room, the layout is shown (L), with some observations (O), ground-truth Q-frames (G) and predicted Q-frames (maximized over the action dimension) (P). Right: Mean squared error (MSE) between the predictions and ground truth, for Q-maps using the proposed convolutional architecture and a simple multilayer perceptron (MLP) in millions of transitions.

## 2.3 EXPERIMENTS

In order to measure the accuracy of the Q-map's representation of the minimum number of steps towards goals and to test its capacity to generalize to a variety of states, we created a simple grid world environment with three different arrangement of obstacles. The environment consists of a simple pixel that can be moved in cardinal directions in a $28 \times 400$ terrain with surrounding walls and non-traversable obstacles of various shapes. When an action would result in colliding with a wall or obstacle, the pixel is kept fixed. The observations consist of a a stack of $28 \times 32$ RGB frames corresponding to the view in a sliding window that moves horizontally with the pixel to keep it in the central column. Walls and obstacles are represented in blue while the pixel is represented in yellow. The first and second rooms share the same obstacle shapes with different locations while the third room consists of drastically different shapes of obstacles.

The Q-map is trained with transitions generated by randomly starting from any free locations in the first room and taking a random action. As a baseline, we used a multi-layer perceptron (MLP) consisting of three fully-connected layers with hidden layers of $1024$ units and layer normalization (Ba et al., 2016) and two output branches consisting of one fully-connected layer for each. The two outputs are aggregated in accordance to the dueling network architecture to represent the final Q-values (Wang et al., 2015), reshaped to $28 \times 28 \times 4$ to create the Q-frames. The proposed convolutional autoencoder-like architecture consists of three layers of convolutions, with dimensions $32$, $32$ and $64$, kernel sizes $8$, $6$ and $4$, and strides $2$, $2$, and $1$ respectively. The convolutions are followed by two fully-connected layers with $1024$ hidden units and layer normalization, followed by two dueling branches consisting of three transposed convolutions (also known as deconvolutions), with dimensions $64$, $32$ and $32$, kernel sizes $4$, $6$ and $4$, and strides $1$, $2$, $2$ respectively. Both networks use exponential linear activation functions.

To evaluate the generalization of the Q-frames we limit the training experience of the agent only to the first room, while evaluating the output against a generated ground-truth Q-frames, generated with path-finding, on all three rooms separately. As can be seen in Figure 2 the Q-frames generated by the convolutional network are almost identical to the ground truth. The agent generalizes extremely well on the second room, which consists of familiar shapes in different arrangements. To our own surprise, the Q-map was able to handle unfamiliar shapes used in the third room to a good

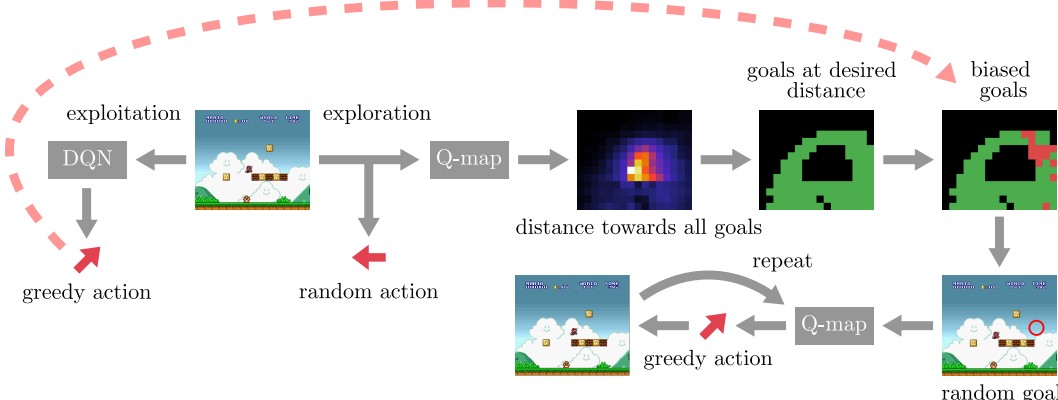

Figure 3: Illustration of the action-decision pipeline of a DQN + Q-map agent.

degree. Overall, the convolutional network has demonstrated significantly faster learning and better generalization properties when compared to the MLP baseline, validating the hypothesis that the proposed convolutional architecture is well suited for the studied scenario.

While the gird world environment is useful for the evaluation of the Q-map performance versus the ground-truth, it does not offer a demonstration of the true capacity of the Q-map agent to handle much more complex visual input. We thus trained the Q-map on Montezuma's Revenge and Super Mario All-Stars games, with examples of the learned Q-frames shown in Figures 4 and 5.

## 3 EXPLORING THE ENVIRONMENT WITH Q-MAP

### 3.1 BACKGROUND

The simplest, but yet very popular, method to explore in RL called $\varepsilon$-greedy, relies on a probability $\varepsilon$ to take a completely random action at any time instead of the best-guess greedy action determined by the agent. The parameter $\varepsilon$ is usually decayed linearly from 1 to a small value through the training to ensure a smooth transition from complete exploration to almost complete exploitation. A major issue is that it relies on random walks to discover new transitions which can generate many repetitive and unnecessary or even destructive transitions. This approach can potentially fail to efficiently push out the exploration boundaries and discover the reward signal required for training.

Several methods have been described that bias the agent's actions towards novelty, primarily by using optimistic initialization or intrinsic rewards (Oudeyer et al., 2007; Oudeyer and Kaplan, 2009; Schmidhuber, 1991; 2010). While the first approach is mainly compatible with tabular methods, the second usually relies on non-environmental rewards provided to the agent, for example, information gain (Kearns and Koller, 1999; Brafman and Tennenholtz, 2002), state visitation counts (Bellemare et al., 2016; Tang et al., 2017) or prediction error (Stadie et al., 2015; Pathak et al., 2017). While these methods show success in sparse reward environments, they dynamically modify the agent's reward function and thus make the environment's MDP appear non-stationary to the agent.

If the agent is goal-oriented it is also possible to explore by providing a range of goals for the agent to reach. For example, in Goal GAN (Florensa et al., 2017) the goals that are at a desired level of difficulty are labeled and used to train a GAN to generate similar goals. Repeating the process enables a curriculum of increasingly difficult goals for the agent to progress through. A main issue with goal-oriented policies used to also learn the primary task, is that the goal representation needs to be sufficiently expressive to represent the required task in the environment, which can make the goal space excessively large and difficult to use.

Finally, RL algorithms that are able to learn off-policy can exploit data collected by a different exploratory agent or provided by demonstration. For example the GEP-PG agent (Colas et al., 2018) uses a policy parameter space search to parametrize an exploratory policy and generate diverse transitions for a DDPG agent (Lillicrap et al., 2015) to learn from, off-policy.

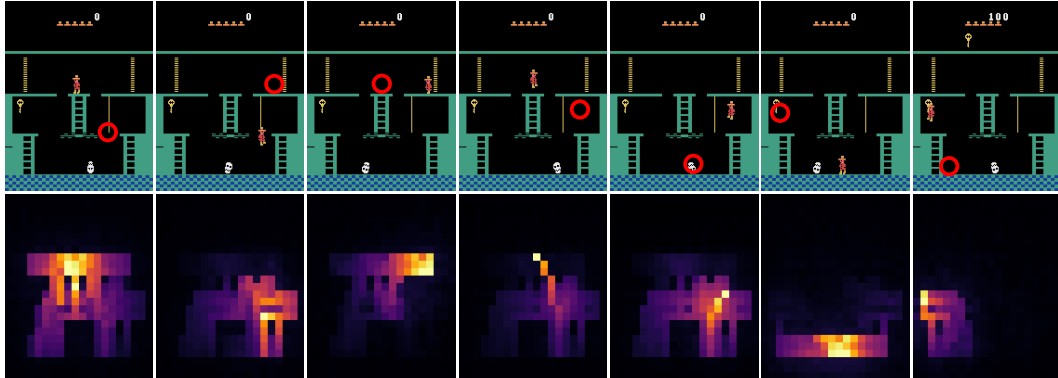

Figure 4: Example of learned Q-frames (maximized over the action dimension) on Montezuma's Revenge and how a simple random-goal walk allows to explore most of the first room. Goals are shown with circles. Most of the room is visited quickly with the proposed exploration method.

## 3.2 THE PROPOSED EXPLORATION ALGORITHM

The main idea of the proposed exploratory algorithm is to replace the random actions in $\varepsilon$-greedy by random goals. The proposed agent uses Q-map to explore and DQN to exploit. At each time step, the action selection follows a decision tree: With probability $\varepsilon_r$ a completely random action can be selected to ensure pure exploration for Q-map and DQN. If no random action has been selected and no goal is currently chosen, a new one is chosen with probability $\varepsilon_g$ by first querying Q-map for the predicted distance towards all the goals, filtering the ones too close or too far away (based on two hyper parameters specifying the minimum and maximum number of steps), choosing one at random and setting a time limit to reach it based on the predicted distance. If a goal is currently chosen, the Q-map is queried to take a greedy action in the direction of the goal and the time allotted to reach it is reduced. Finally, if no random action or goal has been chosen, DQN is queried to take a greedy action with respect to the task.

Because the number of steps spent following goals is not accurately predictable, ensuring that the average proportion of exploratory steps (random and goal-oriented actions) follows a scheduled proportion $\varepsilon$ is not straightforward. To achieve a good approximation, we dynamically adjust the probability of selecting new goals $\varepsilon_g$ by using a running average of the proportion of exploratory steps $\tilde{\varepsilon}$ and increasing or decreasing $\varepsilon_g$ to ensure that $\tilde{\varepsilon}$ approximately matches $\varepsilon$. This allows us later to compare the performance of our proposed agent and a baseline DQN with a similar proportion of exploratory actions.

Unlike with the $\varepsilon$-greedy approach, where an action is either entirely random or greedy, it is possible to bias the exploratory actions towards greedy actions by selecting a goal such that the first action towards the goal is the same as the greedy action proposed by the task-learner agent. Such bias aims to reduce the "cancelling-out" actions usually experienced with random exploration, while still widening the exploration horizon towards the task-learner's intended direction. Finally, it is worth noting that such an exploratory algorithm is entirely compatible with intrinsic reward driven exploration, as such reward can still be provided to the task-learner agent, which would in turn benefit from Q-map trajectories to discover both the intrinsic and environmental rewards.

## 3.3 EXPERIMENTS

For the following experiments, the network used for the Q-map agent is the one described in Section 2.3 with the last convolution's stride set to 2. We use the Montezuma's Revenge environment from OpenAI Gym (Brockman et al., 2016). The observations consist in three $40 \times 56$ grayscale frames created by padding the original frames with zeros to reach a resolution of $160 \times 224$ before scaling with a factor of 4, while the coordinates grid is of size $20 \times 28$. We also use a Super Mario All-Stars environment based on OpenAI Gym Retro with the rewards from the game divided by 100 (0.5 for breaking brick blocks, 1 for killing enemies, 2 for collecting coins, 4 for throwing turtle shells, 10 for eating consumables such as mushrooms, and 50 for reaching the final flag), no bonus for moving to

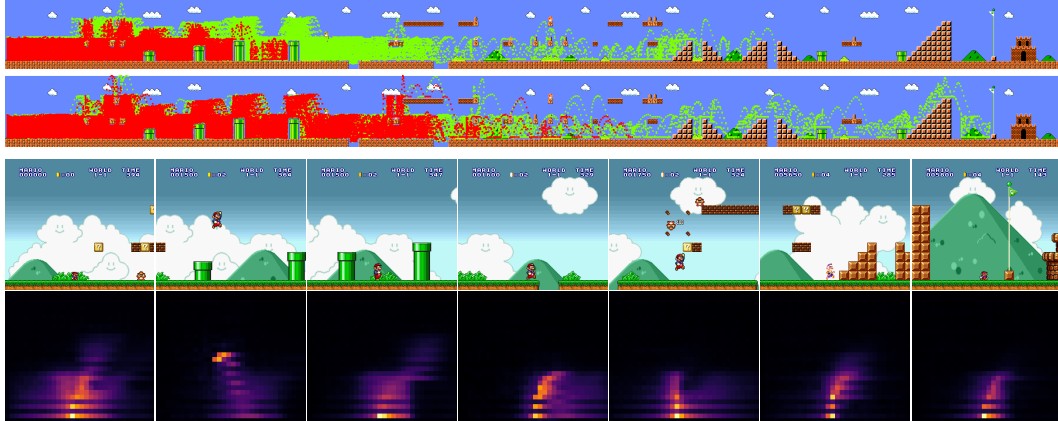

Figure 5: Top: Coordinates visited after 2 million steps on the first level of Super Mario All-Stars. First image: Random walk (red) compared against the proposed Q-map random-goal walk (green). Second image: DQN using $\varepsilon$-greedy (red) compared against DQN with the proposed exploration (green). In both cases, Q-map allows to explore much further. Bottom: Example of learned Q-frames (maximized over the action dimension). The horizontal patterns are due to the action repeat.

the right and no penalty for dying. Terminations by touching enemies or falling into pits are detected from the RAM. The coordinates of Mario and of the scrolling window are extracted from the RAM. Episodes are naturally limited by the timer of 400 seconds present in the game, which corresponds to 2,402 steps. The observations consist in three $56 \times 64$ grayscale frames created with a scaling factor of 4 and the coordinates grid is of size $32 \times 28$. For both environments, we use a simplified action set of size 6 (up-left, up, up-right, left, nothing, right), a frameskip of 8, an action repeat of 4 and a target distance to select goals of 15 to 30 steps. Finally, the time given to reach the goals contains a 50% supplement to account for possible random movements.

For the first experiment we compare the exploration horizons of an agent performing entirely random actions and a Q-map agent. None of them is aware of the environmental rewards or motivated to explore any areas of the environment specifically. We first test both of the agents on the first room of Montezuma's Revenge, as it is a popular and challenging benchmark. We found that the random-action agent was unable to reach the key in 5 million steps while the proposed exploratory agent started to reach it after less than 1.2 million steps and a total of 398 times. Figure 4 shows an exploration episode performed by reaching random goals. As can be seen in Figure 5, the same comparison applied to Super Mario All-Stars revealed that the random agent explores less than 1/3 of the level while the Q-map agent is able to traverse more than 2/3 after 2 million steps.

We then evaluate a combined agent that is comprised of a Q-map exploration agent and a DQN Mnih et al. (2015) task-learner agent. The implementation is based on OpenAI Baselines' DQN implementation Dhariwal et al. (2017), using TensorFlow Abadi et al. (2016). We used double Q-learning Hasselt (2010) with target network updates every 1000 steps, double Q-learning, prioritized experience replay Schaul et al. (2015b) (with default parameters from Baselines) using a shared buffer of 500,000 steps but separate priorities. The training starts after 1000 steps and the networks are used after 2000 steps. Both networks are trained every 4 steps with independent batches of size 32. Two Adam optimizers Kingma and Ba (2014) are used with learning rate $10^{-4}$ for DQN and $3 \times 10^{-4}$ for Q-map, and default other hyperparameters from TensorFlow's implemetation. The probability of greedy actions from DQN increases from 0 to 0.95 linearly throughout the experiment. The Q-map sub-agent outputs 6 frames of size $32 \times 28$ totalling 5,376 Q-values. The goal selection is biased towards the DQN's greedy action with a probability of 0.5 and the discount factor used for Q-map is 0.9 while the task's discount factor is 0.99. Finally, the probability of taking completely random action at any time is decayed from 0.1 to 0.05 throughout the length of the experiment.

To measure the performance of the proposed agent in terms of sum of the rewards collected per episode and number of flags reached we trained it and the baseline DQN for 5 million time steps with 4 different seeds (0, 1, 2, 3) and reported the results in Figure 6. Initially, the combined agent has worse performance than the DQN baseline because random actions result in the agent gathering some

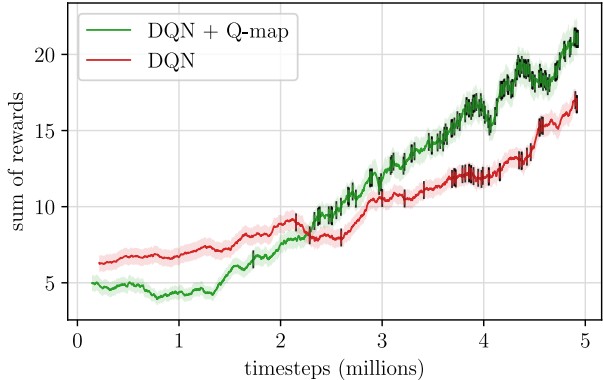

Figure 6: Performance comparison between $\varepsilon$-greedy exploration (red), and the proposed exploration (green) with confidence intervals of 99%. Vertical bars indicate flags reached (end of the level). The proposed agent significantly outperforms the baseline, reaching earlier and more frequently the flag.

early points easily in the first level. However, due to the shorter exploration horizon it takes much longer for the baseline to learn how to progress in the level and at 2 million steps the combined agent overtakes DQN in performance. The final performance of the combined agent is 30% better than the baseline with 33 flags reached on average per seed, versus 9 for the baseline. Finally, we also tested the capacity of the Q-map agent to adapt to a different level and found that the learning was much faster when transferring a pre-trained Q-map from one level to another than when training from scratch as visible in the videos on the website.

## 4    DISCUSSION

A major limitation of the proposed Q-map agent is the requirement to use coordinates as goals and to be able to track the agent's location. However, for compatible environments, Q-map offers a fast and efficient way to learn goal-oriented policies. It is worth noting that the approach could be extended beyond images in observations, such as angles and velocities of objects or point clouds and Q-frames could be replaced by Q-tensors to represent larger coordinate spaces architecturally achieved by using multi-dimensional transposed convolutions. This could, for example, enable the agent to handle scenarios involving robotic arms or flying drones.

In this article we illustrated how Q-maps can be used to improve $\varepsilon$-greedy exploration, but the range of applications is much larger. Any scenario where learning to reach coordinates is useful can benefit from a Q-map approach. For example hierarchical reinforcement learning could combine a high-level agent rewarded by the environment to output coordinates to reach for a low-level Q-map agent while a count-based exploration method could be combined with the estimated distance from a Q-map agent to select close and less visited coordinates.

## 5    CONCLUSION

We proposed a novel reinforcement learning agent that efficiently learns to reach all possible screen coordinates in games. Each past transition can be used to update simultaneously every predictions at once using Q-learning while the convolutional autoencoder-like architecture allows the agent to learn correlations between visual patterns and ways to navigate the environment. We showed that the generated Q-frames match the ground truth and that the agent generalises well on a grid world environment. Furthermore, we proposed a novel exploration method to replace $\varepsilon$-greedy, based on the proposed Q-map agent, and demonstrated that it successfully manages to expand the exploration horizon on Montezuma's Revenge and Super Mario All-Stars allowing to significantly increase the performance of DQN.

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
