# OpenReview forum: "Q-map: a Convolutional Approach for Goal-Oriented Reinforcement Learning"
_ICLR.cc/2019/Conference_

### Official Review · AnonReviewer1 · 2018-11-01
**Do not have enough comparison to existing works; need to improve writing**

**Rating:** 4
**Confidence:** 3

**Review:**

Focus on navigation problems, this paper proposes Q-map, a neural network that estimates the number of steps (in terms of the discount factor gamma) required to reach any position on the observable screen/window. Moreover, it is shown that Q-map can be applied for exploration, by trying to reach randomly selected goal.

Pros
1. Novel goal-based exploration scheme

Cons
1. Similar idea has been proposed before
For example, Dayan (1993) estimates the number of steps to reach any position on the map using successor representations. Discussion about this field (successor representations/features) is completely missing in the paper.
Ref:
- Peter Dayan. Improving generalization for temporal difference learning: The successor representation. Neural Computation, 5(4):613–624, 1993.
- Andre Barreto, Will Dabney, Remi Munos, Jonathan J Hunt, Tom Schaul, David Silver, and Hado van Hasselt. Successor features for transfer in reinforcement learning. In Advances in Neural Information Processing Systems, pp. 4058–4068, 2017.
- Andre Barreto, Diana Borsa, John Quan, Tom Schaul, David Silver, Matteo Hessel, Daniel Mankowitz, Augustin Zidek, and Remi Munos. Transfer in deep reinforcement learning using successor features and generalised policy improvement. In International Conference on Machine Learning, pp. 510–519, 2018.

2. Comparison to existing methods is only vaguely discussed
For example, it is claimed multiple times that UVFA requires the goal coordinates, but Q-map also requires coordinates when doing the exploration.

3. The network architecture is not clearly presented
For example, the output of the network needs to be clipped, which suggests that there is no output transform. Since the predicted output is in [0,1], it would make sense to use Sigmoid transform for each pixel and use logistic loss.

4. The proposed exploration scheme could be unnecessarily complicated
Sec.3.1 provides lengthy discussion about the drawback of eps-greedy exploration. Then in Sec.3.2, \epsilon_r is basically the same as the eps-greedy algorithm, using to randomly select an action. Isn't this a "bad" thing as suggested in Sec.3.1? Moreover, the new exploration scheme requires two more hyper-parameters (min/max distance threshold), which will add more complication to the already very complicated deep RL learning procedure.

5. Experiment results are limited
For the toy experiment in Sec.2.3, the map are relatively simple. The example of Dayan (1993) with an agent surrounded by walls is an interesting scenario and should be included. The proposed Q-map (ConvNet) could fail because it is hard to learn geodesic distance with only local information. More importantly, there is no comparison to similar methods in Sec.3. UVFA can replace Q-map to do similar exploration.

6. Writing can be greatly improved
There are many grammar errors. To name a few, "agent capable to produce", "the gridworld consist of", "in the thrist level".

Minors
- UFV should be UVF in the introduction
- Citation in Sec.3 is not consistent with the rest of the paper. Use \citep or \citet properly.

---

> ### Author Response · Authors · 2018-11-26
> **Response to AnonReviewer1**
>
> 1. Successor features, a generalization of Dayan’s successor representation, propose a framework for transfer learning when the reward function changes between tasks but not the environment’s dynamics. In Dyan (1993), the experiment shows how the successor representations predict the future state occupancy under the current policy when trained to reach a particular goal and describes how the learning is affected when the goal location is changed. We believe this literature is quite different from Q-map which directly and simultaneously learn how to reach every possible goals and is task-independent.
>
> 2. While UVFA requires a goal to be provided in input of the neural network, Q-map doesn't as it produces the Q-values towards all possible goals at once in output. This implies a few algorithmic differences between the two approaches when used for the proposed exploration: 1) During the goal-selection step or training, the values for all goals are queried or updated in one pass through the network while would require as many passes as there are goals with UVFA. 2) When trying to reach a given goal, the Q-values at the proper location in output are used for Q-map while this goal would just be provided in input for UVFA.
>
> 3. We have tested regression with various non-linearities in output but have found them to perform worse. For example, sigmoids tend to squeeze values to either 0 (the goal can't be reached) or 1 (the goal can be reached in one step). Furthermore, clipping is only performed when creating the target Q-frames as always clipping the output of the network would not give any gradient for values outside of (0, 1).
>
> 4. We have retained a minimal amount of purely random actions for several reasons: 1) They are necessary for Q-map's own exploration 2) They allow DQN to discover actions which may not be helpful for navigating the environment, such as hitting blocks to gain coins in Mario 3) The proportion of random actions used is significantly smaller than what is used in the baseline, thus the drawbacks, such as “wasteful” actions, are reduced.
>
> 5. We agree that such environments would have been an interesting test for the Q-map. Given the time available for the rebuttal we will have to consider these for future work. Yes, UVFA could be used instead of Q-map, but it would likely be computationally slower as every possible goal would need to be passed in input and have worse learning performance due to the lack of deconvolutional architecture to facilitate generalization. Such a comparison could also be worthwhile for future work.

---

> > ### Comment · AnonReviewer1 · 2018-12-04
> > **Response**
> >
> > - About UVFA. (a) During training, UVFA does not require querying all goals. As a matter of fact, the whole point of UVFA is to train on a small subset of goals, then to generalize by using the learned neural network. (b) As long as we can query UVFA "the proper location", we can construct similar exploration strategy. Therefore, it would be essential to compare to UVFA in the experiments.
> >
> > - It is also noticeable that the proposed method here is not applicable to continuous state/action space.
> >
> > - If sigmoid + logistic loss performs worse, it would be important to include such experiment (at least in the appendix) to justify your current choices.
> >
> > - Still, the necessity of including \epsilon_r, even though it is smaller than usual \epsilon, implies that the proposed exploration scheme alone is not sufficient and not effective enough.
> >
> > To summarize, the submission is below satisfactory and not ready for publication.

---

### Official Review · AnonReviewer3 · 2018-11-01

**Rating:** 4
**Confidence:** 5

**Review:**

The main idea in the paper is to use on-screen locations as goals for an RL agent. Using a de-convolutional network to parameterize the Q-function allows all goals to be updated at once and correlations between nearby or similar goal locations could be modelled. The paper explores how this type of goal space can be used for better exploration showing modest improvement in scores on Super Mario.

Clarity - The paper is well written and easy to follow. The Q-map architecture is well motivated and intuitive and the exploration strategy based on Q-maps is interesting.

Novelty - The idea of using spatial goals combined with a de-convolutional architecture is not new and goes back at least to “Reinforcement Learning with Unsupervised Auxiliary Tasks” by Jaderberg et al.. The UNREAL agent used the same type of de-convolutional “Q-map” to update a spatial grid of goals all at once. The main difference is that the UNREAL agent learns about spatial goals as an auxiliary task and does not execute/act on the goals like the Q-map agent. Nevertheless, the type of architecture and algorithm (called 3D Q-learning in this paper) is essentially the same.

Significance - The Q-map architecture requires access to the position of the avatar on the screen at training time. I would expect that using such a significant part of the agent’s true state during training should lead to a significant improvement in performance at test time. Why not evaluate the proposed exploration strategy on well known hard exploration tasks? The results on Montezuma’s Revenge are only qualitative. There Q-map agent did outperform an epsilon-greedy DQN baseline on Super Mario but the improvement does not seem very significant given how much prior knowledge Q-map was given compared to the baseline. It is also not clear how much of the improvement comes from training the Q-map as an auxiliary task and how much of it comes from better exploration.

Overall quality - Given that the architecture is not very novel and requires the avatar’s position to train I did not find the qualitative or quantitative results compelling enough. Perhaps the authors could show that the exploration strategy works well on several difficult exploration games. Another possibility would be to showcase other ways to use the Q-map, for example in an HRL setup.

Minor comment - Some sections seem to be missing references. For example, the second paragraph of the introduction discusses GVFs and the Horde architecture without any references.

---

> ### Author Response · Authors · 2018-11-26
> **Response to AnonReviewer3**
>
> We agree with some of the pointed similarities with UNREAL, and now reference it in the paper. The autoencoder architecture and Q-learning used for its pixel-control auxiliary task are indeed similar. However, the meaning of the review's use of the term "spatial goals" is not very clear to us, as the pixel-control auxiliary task's purpose is to maximize the on-screen pixel value change, and has no notion of goal-oriented RL. Furthermore, the learned values are not used in any practical manner. Q-map on the other hand, is trained to minimize the number of steps towards all goal coordinates which can be used for a variety of applications, such as exploration as shown in the paper, goal-oriented control (e.g. if the task is to reach some coordinates), or hierarchical RL.
>
> While we agree that the necessity to localize the agent or a target object in the environment is significant, we would like to point out that it is a common assumption in goal-oriented RL, and is not impractical for certain areas of research, such as robotics. We chose to use Montezuma’s Revenge and Mario for their complexity and their role in various previous papers on exploration. We do not believe it was worthwhile showing performance chart for Montezuma’s Revenge, as the baseline random exploration never reached the key and we did not use environmental rewards.

---

> > ### Comment · AnonReviewer3 · 2018-12-03
> > **Response**
> >
> > The authors agree that the idea of using a deconvolutional architecture for spatially correlated rewards/goals is not new but argue that there is no notion of goal oriented RL in the UNREAL paper. I'm not sure where the difference is since both UNREAL and Q-map learn a set of Q-functions/policies. It is true that UNREAL did not use the auxiliary policies for acting but it is not a stretch to do that. In fact, there is follow up work by Dilokthanakul et al. doing just that (see "Feature Control as Intrinsic Motivation for Hierarchical Reinforcement Learning" - https://arxiv.org/abs/1705.06769).
> >
> > Given that the architecture and training procedure in this paper are not really new, I would expect a really compelling demonstration of how the additional prior knowledge can be used, and I don't think the paper provides that. As I mentioned in the original review, claims of better exploration are usually backed up with experiments on known hard exploration tasks. I would argue that this is even more important for a method that requires additional prior knowledge. If the motivation for the prior knowledge is robotics, then why not evaluate it on a related task?
> >
> > Ultimately, I appreciate the minor revision but stand by my original assessment.

---

### Official Review · AnonReviewer2 · 2018-11-02
**Interesting Idea, but not well evaluated**

**Rating:** 5
**Confidence:** 4

**Review:**

Authors propose to overcome the sparse reward problem using an exploration strategy that incentivizes the agent to visit different parts of the game screen. This is done by building Q-maps, a 3D tensor that measures the value of the agent's current state (defined as the position of the agent) and action in reaching other (x, y) locations in the map. Each 2D slice of the Q-map measures the value at different (x, y) locations for one action. Such 2D slices (i.e. channels) are stacked together to form the Q-map. Taking the max across the channels, thus, provides the Q-value for the optimal action.

A policy for maximizing the rewards is trained using DQN. The Q-map based exploration is used as a replacement for \epsilon-greedy exploration.

The Q-map is used for exploration in the following way:
(a) Chose a random action with probability \epsilon_r.
(b) If neither a random action nor a "goal" is chosen, a new goal is chosen with probability \epislon_g. The goal is a (x, y) location, chosen so that is not too hard or too easy to reach it (i.e. Q-map values are neither too high or low; intuitively [1 - Q-map(x, y, a)] (for normalized/clipped Q) is a measure of distance of the goal).
     -- If a "goal" is chosen, the greedy action to go towards the goal is chosen.
(c) If neither a goal or random action is chosen, DQN is used to chose the greedy exploration.

Authors also bias the goal selection to match DQN's greedy action. This is done as following -- from a set of goals that satisfy (b) above; chose the goal for which Q-map selected action matches the DQN's greedy action.

Results are presented on simple 2D maze environments, Mario and Montezuma's revenge.

I have multiple concerns with the papers:
(i) The writing is informal and the ideas are not well explained. It would really benefit -- if authors introduce an algorithm box or talk about the method as a sequence of points. Right now, the ideas are scattered throughout the paper. I am still confused by figure 3 -- when are random goals chosen? Do random goals correspond to (b) above? Also, when the Horde architecture, GVF and UVF are mentioned, the references are missing -- I would love for the authors to include the corresponding  references.

(ii) The idea of reaching as many states as possible has been explored in count based visitation (Bellemare et al, Tang et al) — but no comparisons have been made to any previous work. Its always good to put a new work in the perspective of old work with similar ideas.

(iii) The authors propose biased and random goal sampling — I would love to see how much improvement does biased goal sampling offer over random goal sampling.

(iv) “…compare the performance of our proposed agent and a baseline DQN with a similar proportion of exploratory actions” .. I don’t agree with this a metric — I think the total number of steps is a good metric. Exploration is part of the agent’s algorithm to find the goal, we shouldn’t compare against DQN by matching the number of exploratory actions.

(v) “The Q-map is trained with transitions generated by randomly starting from any free locations in the environment and taking a random action.” Does this mean that when the agent is trained with Mario — the game is reset after every episode and the agent is placed a random starting location? If yes, then this is not a realistic assumption.

(vi) I would like to see — how do Q-maps generalize across levels of Mario or Montezuma’s revenge? Does Q-map trained on level-1 help in good exploration on future levels without any further fine-tuning?

Overall, I like the idea of incentivizing exploration without changing the reward function as is done in multiple prior works. However, I think more thorough quantitative evaluation is required and it will be interesting to see transfer of Q-maps outside the 2D-domains. I am happy to increase my score if such evidence is provided.

Other references worth including:
(a) Strategies for goal generation: Automatic Goal Generation for Reinforcement Learning Agents (https://arxiv.org/abs/1705.06366)

---

> ### Author Response · Authors · 2018-11-26
> **Response to AnonReviewer2**
>
> First, we would like to clarify that this paper makes two main contributions: 1) Q-map: a way to simultaneously learn to reach coordinates and 2) DQN + Q-map: a way to use Q-map for exploration. Unfortunately the review’s points did not address 1).
>
> (ii) We do reference these works in section 3.1, however, as most of them still use epsilon-greedy as part of their algorithms, our proposed method can be directly integrated with them. To isolate the impact of taking multiple steps in the direction of a goal versus random actions, we chose to only use a standard DQN agent.
>
> (iii) We unfortunately do not have results with the exact same experimental setup without goal biasing but during preliminary experiments we found that a goal biasing of 50% gave a performance boost on Mario. The experiment with Montezuma's Revenge does not use any biasing however as no reward was used and thus no DQN was trained.
>
> (iv) By exploratory actions we mean individual actions that are not greedy for the task (completely random or goal-directed). To have a fair comparison between epsilon-greedy exploration and the proposed exploration using Q-map, we ensure that these exploratory actions are following the same schedule, linearly decaying through the training.
>
> (v) The quoted training method was specifically used for the gridworld environment that was designed to evaluate the training of the Q-map under ideal conditions with a nearly uniform coverage of all transitions. For the experiments with Mario and Montezuma's Revenge the goal was to evaluate the proposed exploration algorithm, we therefore used the original starting states at the beginning of the levels.
>
> (vi) We added a new experiment using a Q-map trained first on level 1.1 and then on level 2.1. We noticed faster training and some notions of generalization even though the two levels use different tilesets and backgrounds. The videos and code are available on the website.

---

> > ### Comment · AnonReviewer2 · 2018-12-10
> > **Insufficient Experiments**
> >
> > While authors have addressed some concerns, they have not addressed others. I would encourage the authors to conduct thorough experimental evaluation and resubmit the paper.

---

### Author Response · Authors · 2018-11-26
**General comment**

We thank the reviewers for their constructive feedback. It has helped us improve the quality of the paper and gave us directions for future work.

Since the original submission, we have updated the paper and improved the website https://sites.google.com/view/q-map-rl with some new videos and cleaner source code.

---

### Public Comment · ~Shishir_Sharma1 · 2019-01-09
**Findings of the ICLR 2019 reproducibility challenge**

The paper proposes an exploratory algorithm which replaces exploration approach such as e-greedy, which only relies on random walks, in favor of a goal oriented Reinforcement Learning (RL) approach. The authors propose Q-map, a convolutional autoencoder-like architecture that is used to simultaneously produce value estimates for all possible goals in compatible environments i.e environments that support spatial coordinates as goals. Finally, the authors report the results of a RL agent that explores using Q-map and exploits using a DQN on Montezuma’s Revenge and Super Mario All-Stars environment.

We tried to reproduce the authors' result on the Super Mario All-Stars environment and sought to extend the scope of the experimentation by testing the generalization of the agent as well. The authors have made public their code and we ported it to the Pytorch framework. While the authors have mentioned the details of most of the hyperparameters being used, the details regarding how the authors deal with a sliding window present in the environment are a little unclear.

The results regarding the comparison of the performance of the proposed algorithm and the baseline that we generated differ from those in the paper. Our results indicate a better performance of the baseline algorithm than the proposed algorithm. We tried to recreate the exact conditions under which the results in the paper were observed but the paper mentions an averaging of multiple runs of the algorithm with different seeds. As the algorithm and the baseline are required to be run for 5 million timesteps, which together end up taking 6 days to complete, we had to report the results for only a single run. Without any training or finetuning, the proposed agent generalizes poorly on unseen level, though this can be explained by the very different backgrounds of the levels.

---

### Meta-Review · Area_Chair1 · 2018-12-13
**Promising but more thorough investigation needed**

**Confidence:** 5
**Recommendation:** Reject

**Metareview:**

The paper proposes to use a convolutional/de-convolutional Q function over on-screen goal locations, and applied to the problem of structured exploration. Reviewers pointed out the similarity to the UNREAL architecture, the difference being that the auxiliary Q functions learned are actually used to act in this case.

Reviewers raised concerns regarding novelty, the formality of the writing, a lack of comparisons to other exploration methods, and the need for ground truth about the sprite location at training time. A minor revision to the text was made, but the reviewers did not feel their main criticisms were addressed. While the method shows promise, given that the authors acknowledge that the method is somewhat incremental, a more thorough quantitative and ablative study would be necessary in order to recommend acceptance.